# The Impact of a ‘Remotely-Delivered’ Sports Nutrition Education Program on Dietary Intake and Nutrition Knowledge of Junior Elite Triathletes

**DOI:** 10.3390/nu14245203

**Published:** 2022-12-07

**Authors:** Xuedan Tan, Natalie Rogers, Nancy Brown, Melanie MacDonald, Amy-Lee Bowler, Gregory R. Cox

**Affiliations:** 1Nutrition and Dietetics, Faculty of Health Sciences and Medicine, Bond University, Robina 4226, Australia; 2Triathlon Australia, Gold Coast 4227, Australia

**Keywords:** education, endurance athletes, triathlon, carbohydrate intake, nutritional analysis

## Abstract

Triathlon is a physically demanding sport, requiring athletes to make informed decisions regarding their daily food and fluid intake to align with daily training. With an increase in uptake for online learning, remotely delivered education programs offer an opportunity to improve nutritional knowledge and subsequent dietary intake in athletes. This single-arm observational study aimed to evaluate the effectiveness of a remotely delivered nutrition education program on sports nutrition knowledge and the dietary intake of junior elite triathletes (*n* = 21; female *n* = 9; male *n* = 12; 18.9 ± 1.6 y). A total of 18 participants completed dietary intake assessments (4-day food diary via Easy Diet Diary^TM^) and 14 participants completed an 83-question sports nutrition knowledge assessment (Sports Nutrition Knowledge Questionnaire (SNKQ)) before and after the 8-week program. Sports nutrition knowledge scores improved by 15% (*p* < 0.001, ES = 0.9) following the program. Male participants reported higher energy intakes before (3348 kJ, 95% CI: 117–6579; *p* = 0.043) and after (3644 kJ, 95% CI: 451–6836; *p* = 0.028) the program compared to females. Carbohydrate intake at breakfast (*p* = 0.022), daily intakes of fruit (*p* = 0.033), dairy (*p* = 0.01) and calcium (*p* = 0.029) increased following nutrition education. Irrespective of gender, participants had higher intakes of energy (*p* < 0.001), carbohydrate (*p* = 0.001), protein (*p* = 0.007), and fat (*p* = 0.007) on heavy training days compared to lighter training days before and after the program with total nutrition knowledge scores negatively correlated with discretionary food intake (*r* = −0.695, *p* = 0.001). A remotely delivered nutrition education program by an accredited sports nutrition professional improved sports nutrition knowledge and subsequent dietary intake of junior elite triathletes, suggesting remote delivery of nutrition education may prove effective when social distancing requirements prevent face-to-face opportunities.

## 1. Introduction

Triathlon is a physically demanding sport, involving between 20–30 h of training per week across three disciplines, swimming, cycling, and running [1]. Junior triathletes face several challenges when attempting to meet increased nutritional requirements associated with growth and development in addition to high daily exercise loads. Commonly reported barriers experienced among triathletes include lack of nutritional knowledge [2], limited time available to eat, and high food costs [3,4]. Further, most junior athletes have university or school commitments, are required to travel for daily training and competition, lack proficient food preparation skills and are susceptible to being influenced by family and friends. Given that inadequate nutritional intakes and inappropriate fluid intake strategies are associated with impaired performance, delayed growth and increased risk of injury [5], young elite triathletes require sound nutrition knowledge and skills to ensure appropriate manipulation of daily food and fluid intakes to reflect fluctuations in daily and seasonal training load. 

Although the importance of adequate nutrition knowledge for optimal dietary intake is recognized, the recommendations for post-exercise carbohydrate and protein intakes are not well understood among triathletes [6]. Often, triathletes consume inadequate energy intake to support daily energy expenditure and have been reported to consume inadequate dietary intakes of iron and calcium [7,8]. To address these concerns, nutrition education programs designed to assist athletes in acquiring nutrition knowledge and improving dietary behaviors should be implemented to provide triathletes with the necessary skills to manipulate daily energy and nutrient demands to reflect daily training loads [9]. 

While sports nutrition education interventions typically improve nutrition knowledge and eating habits among athletes in team sports [10], the current understanding of dietary changes associated with nutrition education interventions in endurance athletes is limited [11]. Further, the effectiveness of remotely delivered nutrition education to improve nutrition knowledge and subsequent dietary intake of athletes is mostly unknown, particularly in young elite endurance athletes [9,12]. Given the social distancing requirements elicited by the COVID-19 pandemic and the subsequent uptake of online education, Triathlon Australia facilitated a remotely delivered nutrition education program to support performance, health and well-being of junior elite triathletes. This study aimed to evaluate the effectiveness of a remotely delivered nutrition education program on nutrition knowledge and subsequent dietary intake of junior elite triathletes.

## 2. Materials and Methods

### 2.1. Study Design and Participants

Twenty-five triathletes attending the Triathlon Australia Performance Health Program (TAPHP) were approached by the primary researcher (GC) to participate in this study. The TAPHP was a remotely delivered 8-week program that included a medical review and subsequent follow-up, alongside weekly nutrition education sessions. Verbal and written communication of the scope and risks of the study were provided to participants and their guardians prior to their involvement. Twenty-one triathletes were recruited (female *n* = 9; male *n* = 12; 18.9 ± 1.6 y) and provided written informed consent, including guardian consent for those aged under 18 years. Participants who failed to complete the dietary intake and nutrition knowledge assessments were excluded from final analysis. Permission to conduct the observational study was granted by the relevant ethics board at Bond University, Australia (BUHREC; no. GC02696). 

### 2.2. Study Overview 

The TAPHP was developed to educate triathletes on the importance of nutrition to promote health, well-being, and performance when undertaking high daily training loads. The nutrition education program was conducted over 8 weeks and consisted of (1) a remotely delivered 5-week nutrition education series (Weeks 1–5; Microsoft Teams, Version 4.9.12.0); and (2) a 30-min individual dietary consultation (Weeks 6–8) delivered by an Accredited Sports Dietitian (MM). The weekly nutrition education topics included: training nutrition, energy availability, hydration, macronutrient (energy, carbohydrate, protein, and fat) and micronutrient (iron and calcium) requirements, supplements, food safety, and travel nutrition. Visual aids, online chat (Microsoft teams, Version 4.9.12.0) and short quizzes (Mentimeter, Version 3.2.5, Sweden) were used to engage athletes during sessions. Several practical activities were also used, including meal planning for heavy training days and snack/smoothie preparation. Relevant nutrition factsheets were developed and provided electronically to the athletes immediately after each session. Immediately before and after the program, participants completed an 83-question Sports Nutrition Knowledge Questionnaire (SNKQ) and a 4-day food diary (Easy Diet Diary, Xyris Software Pty Ltd., Brisbane, Australia, 2019).

### 2.3. Sports Nutrition Knowledge Questionnaire (SNKQ)

An online Sports Nutrition Knowledge Questionnaire (SNKQ) was administered to participants immediately before and after the 8-week nutrition education program via Qualtrics Experience Management (Qualtrics XM) Survey Software (Qualtrics, Drive Provo, UT, USA). The SNKQ, which has been previously validated to assess the nutrition knowledge of athletes [13], was modified by an experienced Accredited Sports Dietitian (GRC) who has more than 20 years’ experience in triathlons to ensure the questions were relevant to the participants. The questionnaire included 83 multiple-choice style questions divided across two parts: (1) participant demographics; and (2) sports nutrition knowledge that covered four themes (nutrients, hydration, training nutrition and supplements).

Participants received a score of +1 for a correct answer and 0 if the selected response was incorrect or left blank. The scores for each subsection and total score were calculated based on the number of questions answered correctly with a maximum total score of 83.

### 2.4. Dietary Intake Assessment 

Dietary intake assessments using a 4-day food diary (two weekdays and two weekend days) in the mobile application Easy Diet Diary™ (Xyris Software Pty Ltd., Australia, 2019) were completed before and after the nutrition education program. Participants used household measures to estimate intake and were asked to photograph and record timing of meals and snacks to allow for greater accuracy in reporting. Participants were advised to record the timing, duration (mins), distance (km) and sports discipline (swimming, cycling, running, strength training) of all training sessions during the 4-day collection period. Dietary intake data was tracked in real-time via Easy Diet Diary Connect (Xyris Software Pty Ltd., Australia, 2019) and members of the research team (Accredited Sports Dietitians (MM and AB)) prompted participants individually throughout the collection period to minimize recorder error and ensure accurate dietary collection. Participants were sent daily reminders during the recording period and were supported with expert assistance (MM and AB) to ensure recorded food diaries reflected actual dietary intakes.

Dietary intake data was entered into FoodWorks™ (Xyris Software, FoodWorks 8 Professional, Australia, 2015) for final nutrient analysis and cross-checked for entry errors (NR and GC). Energy, macronutrient (carbohydrate, fat, protein, alcohol), and micronutrient breakdowns (iron and calcium) were calculated for each participant before and after the nutrition education program. Dietary collection days were categorized as light (up to one single training session ≤ 1 h per day) and heavy (a single session > 1 h or multiple training sessions). Dietary intake was categorized according to meal timing (breakfast: 0500–0859; lunch: 1100–1459; dinner: 1700–2059; and snacks: food and fluid consumed outside of mealtimes and/or before, during or after training). Dietary quality was assessed by comparing dietary intake to the Australian Guide to Healthy Eating (AGHE) [14]. Foods and fluids were itemized into the following groups—fruits, vegetables, meats and alternatives, bread and cereals, dairy and alternatives and discretionary. Discretionary foods are low-nutrient quality foods high in saturated fat and/or added sugars, added salt or alcohol and low in fiber. Sports foods, such as sports drinks, protein supplements, and sports gels, were classified as a separate food group according to the companies’ serve sizes.

### 2.5. Statistical Analysis 

All statistical analyses were conducted using the Statistical Package for Social Sciences (IBM SPSS Statistics for Windows, Version 26.0), where appropriate, results are reported as mean ± SD (normally distributed continuous data) or median and IQR (non-normally distributed continuous data). Categorical data are presented as percentage of total responses n (%). Shapiro–Wilk tests were used to determine normality of data and Wilcoxon Signed Ranks Tests performed for non-normally distributed variables. Paired sample *t*-tests were used to determine mean difference between pre- and post-program nutrition knowledge scores and compare dietary intake on light vs. heavy training days, pre- vs. post-eating occasions and weekday vs. weekend intakes. Independent *t*-tests were employed to compare differences in pre- and post-data (nutrition knowledge and dietary intake) between females and males. One-way ANOVAs were used to assess differences in nutrition knowledge against various participant characteristics such as level of competition, and level of nutrition knowledge at baseline, and reported energy and macronutrient intakes at different meals (Breakfast, Lunch and Dinner). Pearson’s correlation (normally distributed data) or Spearman’s correlation (non-normally distributed data) was used to determine the relationship between nutrition knowledge and dietary intake. Significance was set at *p* ≤ 0.05. The effect sizes (Cohen’s D) for nutrition knowledge scores were calculated and were interpreted as small (0.2–0.4), medium (0.5–0.7), and large (≥0.8) [15]. 

## 3. Results

### 3.1. Participants Characteristics

A total of 18 participants (*n* = 18) completed the 4-day food diary and 14 participants (*n* = 14) completed the SNKQ before and after the program. The demographic characteristics of recruited participants (*n* = 18) are summarized in Table 1. A total of 17 participants (*n* = 17) did not follow a specific diet, while 1 reported following a paleolithic diet and 2 participants reported allergies to lactose and nuts. Participants reported coaches (*n* = 11, 61%), sports dietitians (*n* = 11, 61%), and parents (*n* = 10, 56%) as the three most common sources of pre-existing nutrition information. Secondary sources of nutrition information were the internet (*n* = 7, 39%), school or university (*n* = 5, 28%), doctors or other medical staff (*n* = 4, 22%), and social media (*n* = 1, 10%).

### 3.2. Sports Nutrition Knowledge

SNKQ scores of participants before and after the program (*n* = 14) are outlined in Table 2. There was a significant improvement in total SNKQ score (7.9, 15%; *p <* 0.001; ES = 0.9) and subsection scores for hydration (1.2, 19%; *p <* 0.001; ES = 1.1), training nutrition (2.6, 17%; *p <* 0.001; ES = 1.4) and nutrients (3.4, 12%; *p =* 0.010; ES = 0.6) following the nutrition program. No significant improvement was observed for the supplement subsection *(p* = 0.19). Females had a significantly higher SNKQ total score (8.5%, *p* = 0.031) and nutrient subsection score (5%, *p* = 0.015) at baseline compared to males. Following the program, there was no difference in total SNKQ score or rate of improvement between female and male participants (*p* = 0.397). No differences in pre- or post-program SNKQ total scores were found between the level of competition (*p* = 0.226) or the number of years competing (*p* = 0.255). Pre-program, no differences in SNKQ total score were observed between participants who had previously received sports dietitian education and those without prior education (*p* = 0.639).

### 3.3. Dietary Intake 

At baseline, male participants reported higher daily energy (3348 kJ, 95% CI: 117–6579; *p* = 0.043), carbohydrate (105 g, 95% CI: 11–198; *p* = 0.03), fat (36 g, 95% CI: 1–72; *p* = 0.046) and iron (6 mg, 95% CI: 1–11; *p* = 0.016) intakes compared to females. Similarly, male participants reported higher daily energy (3644 kJ, 95% CI: 451–6836; *p* = 0.028) and fat (40 g, 95% CI: 1–78; *p* = 0.043) intakes compared to females following the education program. Calcium intake was significantly higher after the program among both males and females (254 mg, *p* = 0.029) (Table 3). No other differences were observed in reported energy or nutrient intake pre vs. post program.

Energy (3321 kJ, 95% CI: 1988–4654; *p* < 0.001), carbohydrate (93 g, 95% CI: 47–139; *p* = 0.001), protein (30 g, 95% CI: 10–50; *p* = 0.007) and fat (27 g, 95% CI: 9–46; *p* = 0.007) intakes were higher on heavy compared to light training days both before and after the program. There was no difference in energy, carbohydrate, protein, or fat intake between weekdays and weekends. 

Participants reported higher energy (1035 kJ, 95% CI: 113–1957; *p* = 0.023 and 1322 kJ, 95% CI: 419–2225; *p* = 0.003), protein (23 g, 95% CI: 10–36; *p* < 0.001 and 16 g, 95% CI: 3–30; *p* = 0.015, respectively), and fat (13 g, 95% CI: 0–25; *p* = 0.044 and 18 g, 95% CI: 6–30; *p* = 0.003, respectively) intakes at dinner compared to lunch, before and after the program. Similarly, participants reported higher energy (2351 kJ, 95% CI: 1399–3302; *p* < 0.001 and 1943 kJ, 95% CI: 863–3023; *p* < 0.001, respectively), protein (45 g, 95% CI: 35–55; *p* < 0.001 and 35 g, 95% CI: 22–48; *p* < 0.001, respectively) and fat (31 g, 95% CI: 19–44; *p* < 0.001 and 32 g, 95% CI: 17–46; *p* < 0.001) intakes at dinner compared to breakfast, before and after the program. There was a significant increase in carbohydrate intake at breakfast (17 g, 95% CI: 3–31; *p* = 0.022) following the program. 

Food group intake was assessed using the AGHE for healthy non-athletic populations, with an additional food group added (sports foods) (Table 4). At baseline, 50% (*n* = 9) of participants met their age-appropriate daily fruit intake recommendations, which increased to 72% (*n* = 13) after the program (0.8 ± 0.07 serving difference, *p* = 0.033). Similarly, milk and alternatives increased from 55% (*n* = 10) of participants meeting recommended serves, to 61% (*n* = 11) after the program (0.7 ± 0.2 serving difference, *p* = 0.01). There was no change in intake for other food groups following the program. 

### 3.4. Relationship between Nutrition Knowledge and Dietary Intake

SNKQ total score was negatively correlated with discretionary food intake, irrespective of the program (*r* = −0.695, *p* = 0.001). No significant associations were observed between intake from other food groups and sports nutrition knowledge. There was no correlation between nutrient intake (energy, protein, fat, carbohydrates, iron, and calcium) and SNKQ total score before or after the program.

## 4. Discussion

This study evaluated the effectiveness of a remotely delivered nutrition education program on the sports nutrition knowledge and dietary intake of junior elite triathletes during a period when social distancing prevented face-to-face interactions. The sessions were delivered using interactive online tools and tailored to the specific needs of triathletes, covering topics such as training nutrition, energy availability, hydration practices, macronutrient and micronutrient recommendations, food safety, supplements, and dietary considerations for travelling. The primary findings from this study were (1) sports nutrition knowledge and subsequent dietary intake of junior elite triathletes improved following the nutrition education program; (2) participants reported higher energy, carbohydrate, protein, and fat intakes on heavy training days compared to light training days, irrespective of the education program; and (3) athletes increased their carbohydrate intakes at breakfast following the program reflecting a favorable change in dietary behavior, which emphasizes appropriate fueling ahead of daily training. Despite the challenges the COVID-19 global pandemic has created for the delivery of effective and meaningful interactions between performance support staff and athletes, the findings from this study suggest that remotely delivered nutrition education programs provide an avenue to align dietary intakes of athletes with daily nutrient requirements. 

Junior elite triathletes who completed the 8-week nutrition education program in this study improved sports nutrition knowledge scores by 15%. Previous studies investigating the effectiveness of online education delivery approaches have reported similar improvements in athlete nutrition scores ranging from 6.4–25.2% [16,17,18,19]. The participants in this study had the greatest improvement in hydration and training nutrition subsections scores, which may be explained by a greater focus on hydration practices and strategic training nutrition within the 8-week nutrition education program, given the importance of carbohydrate, protein, and fluid in endurance sports [20,21]. This finding is consistent to a previous study that demonstrated an improvement in sports nutrition knowledge (8.3%) after a 7-week classroom-based nutrition education program among highly trained adolescent swimmers [22]. Alongside the improvement observed in nutrition knowledge scores, athletes increased their intake of foods from fruit and milk and alternatives food groups following the program. The latter dietary change is of particular importance for endurance athletes, given the important role calcium plays in supporting bone health [8]. The increased number of serves from milk and alternatives food groups resulted in an increase in daily calcium intake of ~18%. This represents a favorable change as endurance athletes (i.e., runners and triathletes) undertaking high daily training loads, while maintaining lean physiques are at increased risk of bone stress injuries [23]. While we observed a positive change in nutrition knowledge with increased consumption of fruit and milk and alternative food groups after the nutrition education program, the long-term impact of the program on athletes’ nutrition knowledge and subsequent dietary intake was not explored beyond this period.

We devoted 10 of 83 questions within the sports nutrition knowledge questionnaire to hydration and included questions that assessed knowledge of beverage composition, implications of altered hydration status and appropriate fluid intake strategies for triathlon training specific scenarios. Ad libitum fluid intake during exercise is multifactorial [24] and likely influenced by external cues, such as coach’s encouragement to drink [25] with athletes relying on performance support staff (i.e., athletic trainers) for hydration information [26]. While we carefully monitored dietary intake to assess daily energy and macronutrient intakes, we did not require participants to track daily fluid intake of non-calorie containing fluids and subsequently failed to measure daily total fluid intake before and after the education program. High sweat rates are commonly reported in endurance sport athletes [27], yet daily fluid intakes are poorly understood. Reporting total daily fluid intake and assessing hydration status would have added important additional context to our study and would be a worthy inclusion in future studies assessing changes in daily food and fluid intake in response to nutrition education interventions.

In this study, higher nutrition knowledge scores were associated with lower discretionary (low nutrient quality) food intake, which is consistent with previous research that demonstrated that nutrition knowledge scores were negatively correlated with the consumption of carbonated drinks and processed foods in athletes [28,29]. Heavy reliance on discretionary foods in close proximity to daily training sessions can displace the intake of nutrient dense, carbohydrate and protein rich foods important for post exercise recovery. Reported dietary iron intakes of participants in the current study met the recommended dietary intake (RDI) for males (11 mg) and females (15 mg) [30]. Of interest, only one participant correctly answered the question regarding the absorption ratio of iron in plant and animal foods. As endurance athletes are considered at higher risk of micronutrient deficiencies (i.e., iron) compared to other athlete groups [7], future nutrition education interventions should emphasize the importance of consuming nutrient rich foods, while addressing current sports nutrition recommendations for carbohydrate and protein and provide practical strategies to assist athletes to overcome day-to-day sport-specific scenarios [31].

A novel finding from the present study was that participants aligned their daily dietary intakes with fluctuations in daily training. Elite junior triathletes in this study, reported higher intakes of energy, carbohydrate, protein, and fat on heavy training days compared to light training days before and after the nutrition education program. Similarly, elite adolescent triathletes living in a full-board residential campus have previously been shown to increase relative (9.0 vs. 7.8 g.kg body mass) daily carbohydrate intake during an intensive training week compared to a moderate week [32]. A noticeable difference in this study was that athletes had easy access to ready prepared meals and snacks when eating in a residential campus [32]. However, previous reports suggest that these findings are not universal, as adolescent athletes commonly fail to adjust their nutritional intake to different daily training volumes [33]. Aligning dietary intake with daily training is particularly important for endurance athletes as they are at an increased risk of low energy availability (LEA) [34] and nutrient deficiencies [7,8]. LEA can lead to significant perturbations in reproductive functioning, hormonal control, bone health and immune function, and ultimately impair sporting performance [34]. Indeed, the energy intakes of triathletes in the present study are similar to those previously reported for elite adolescent male triathletes living in a full-board residential campus as described above [32]. Dietary energy intakes observed of elite adolescent triathletes in the present study were higher than the commonly reported intakes of other endurance athletes, whom often fail to meet estimated daily energy expenditures [35]. The higher energy intakes reported in the current study could be explained by participants’ prior experience with individual or group education with an Accredited Sports Dietitian and/or appropriate dietary messaging from coaches and parents. An alternate explanation may be due to the dietary recording process employed in the current study. Participants were provided clear instructions for dietary recording, which was made easier by utilizing a dietary app (Easy Diet Diary™), allowing participants to upload photo images of meals. Further participants were contacted regularly during the dietary recoding period. Participants’ dietary intakes were tracked in real-time throughout the day and any delays in entering data or obvious discrepancies were followed up immediately. Of interest, participants responded promptly to text reminders throughout the recording period but were less likely to respond to verbal follow-up by the research team seeking clarification.

Daily macronutrient intake remained unchanged following the 8-week nutrition education program, however there was an important change in the patterning of intake over the day. While the evening meal (dinner) was higher in energy, carbohydrate, protein, and fat compared to other meals and snacks consumed throughout the day, carbohydrate intake at breakfast increased following the nutrition education program. This change in dietary patterning is important as altering the timing of carbohydrate intake around exercise can optimize the response to daily training [36]. Further, consumption of carbohydrate immediately before exercise represents an effective strategy to provide an exogenous fuel source to the muscle and central nervous system during extended exercise sessions [37]. The observed increase in carbohydrate intake at breakfast suggests that athletes attempted to implement strategies discussed during the education program. A theme addressed during the training nutrition module was to incorporate carbohydrate rich foods early in the day to support recovery from morning training sessions, as well as refuel ahead of sessions scheduled later in the day. Unfortunately, physical characteristics (including height and body mass) of triathletes in the present study were not collected (directly or self-reported). When planning the current study, we were aware of the weight stigma that prevails in endurance sports [38], particularly amongst junior athletes [39]; therefore, we took a conservative approach and deliberately excluded height and body mass when collecting participant demographics. As such, athlete carbohydrate intakes could not be compared directly to sport-specific carbohydrate intake guidelines [31], which does limit the interpretation of our results. Nevertheless, we observed positive changes in daily carbohydrate intake patterning that aligned with focused messages incorporated in the education program. Detailed examination of dietary patterning throughout the day, alongside careful assessments of energy expenditure in future studies would provide a more thorough understanding of within-day energy balance, which would assist in identifying dietary intake adjustments required to align with daily training sessions [40,41]. In future education programs, practitioners should consider the logistical challenges of implementing dietary suggestions around daily training to ensure athletes are able to make meaningful changes in their nutrient intake.

One of the perceived limitations of the current study is the small sample size (*n* = 18); however, this study does include almost all elite junior triathletes categorized by Triathlon Australia (18 of 25 categorized athletes). The aim of this study was to determine the effectiveness of a tailored, remotely delivered education program in emerging triathletes, which future studies can build upon amongst other triathlon cohorts (i.e., masters athletes) who have been previously shown to lack knowledge of sports nutrition guidelines [6]. As previously mentioned, a strength of the current study was the rigor undertaken to collect dietary intake data, however methods to assess daily training load, such as the use of cycling power meters and accelerometers would have strengthened our assessment of daily exercise energy expenditure. However, these methods are costly, burdensome on participants and often impractical in studies involving free-living participants, particularly when participants are located remotely, undertaking training sessions across different exercise modalities as is the case in triathlon (swimming, cycling and running). In addition, we did not include a control group in our study, which prevents firm conclusions regarding the effectiveness of the remotely delivered education program utilized. Further, while we assessed sports nutrition knowledge and dietary intake immediately following the education program, future research should implement follow-up assessments at 6 and 12 months, preferably during standardized training periods, to assess the long-term effectiveness of remotely delivered education programs.

## 5. Conclusions

In conclusion, results from this study suggest that a remotely delivered 8-week nutrition education program by an Accredited Sports Dietitian can improve the nutrition knowledge and dietary intakes of junior elite triathletes. Specifically, athletes improved their nutrition knowledge by 15%, which resulted in positive dietary changes that better aligned dietary intake to daily training. The nutrition education program used in this study provides a template for future education programs when social distancing requirements prevent face-to-face opportunities. Future research is warranted to explore long-term changes in sports nutrition knowledge and subsequent dietary behaviors in response to novel education programs that accommodate social distancing requirements, such as online modules or facilitated education sessions, mobile phone applications or social media-based interventions.

## Figures and Tables

**Table 1 nutrients-14-05203-t001:** Demographic characteristic of elite junior triathletes undertaking the 8-week nutrition education program (*n* = 18).

(Mean ± SD)	
Age	18.9 ± 1.6
Yeas of competing in Triathlon	5 ± 1.8
Weekly training hours	19.2 ± 3.6
Gender [*n*, (%)]	
Female	9 (50%)
Male	9 (50%)
Country of birth [*n*, (%)]	
Australia	16 (88%)
Zimbabwe	1 (6%)
South Africa	1 (6%)
Level of competition [*n*, (%)]	
World Triathlon	8 (45%)
Oceania Triathlon	4 (22%)
National Triathlon	6 (30%)
Previous nutrition education [*n*, (%)]	
Individual sports dietitian consultation	11 (61%)
Group education session	13 (72%)
None	3 (17%)

SD: standard deviation; *n*: number.

**Table 2 nutrients-14-05203-t002:** Sports Nutrition Knowledge Questionnaire (SNKQ) scores of junior elite triathletes (mean ± SD; % correct scores) before and after the 8-week nutrition education program (*n* = 14).

Sections (No. of Questions)	Pre-Program Scores	Post-Program Scores	*p*-Value	Effect Size
Nutrients (43)	27.5 ± 5.7; 64.0%	30.9 ± 6.0; 71.9%	0.010	0.6
Hydration (9)	6.3 ± 1.3; 69.8%	7.5 ± 0.9; 83.3%	0.001	1.1
Training nutrition (24)	15.1 ± 1.8; 62.8%	17.7 ± 2.0; 73.8%	0.001	1.4
Supplements (7)	3.9 ± 1.5; 56.1%	4.6 ± 1.4; 65.3%	0.168	0.4
Total (83)	52.8 ± 8.3; 63.6%	60.7 ± 8.8; 73.1%	0.000	0.9

**Table 3 nutrients-14-05203-t003:** Daily nutrient intakes of junior elite triathletes (mean ± SD) before and after the 8-week nutrition education program (*n* = 18).

Nutrient Intake	Combined PRE (*n* = 18)	Combined POST (*n* = 18)	Male PRE (*n* = 9)	Male POST (*n* = 9)	Female PRE (*n* = 9)	Female POST (*n* = 9)	Light Training Days (*n* = 18)	Heavy Training Days (*n* = 18)
Energy (kJ/day)	14,298 ± 3578	14,436 ± 3622	15,972 ± 2958 ^a^	16,258 ± 2115 ^a^	12,624 ± 3487	12,614 ± 3993	12,708 ± 3232 ^b^	16,028 ± 4553
CHO (g/day)	376 ± 105	393 ± 108	428 ± 79 ^a^	442 ± 78	323 ± 105	344 ± 116	330 ± 107 ^b^	423 ± 136
Protein (g/day)	164 ± 43	157 ± 40	174 ± 46	173 ± 33	154 ± 40	140 ± 41	147 ± 39 ^b^	177 ± 54
Fat (g/day)	131 ± 39	132 ± 43	149 ± 35 ^a^	152 ± 37 ^a^	113 ± 36	112 ± 40	122 ± 30 ^b^	149 ± 51
Calcium(mg/day)	1445 ± 554	1699 ± 560 *	1689 ± 589	1924 ± 556	1201 ± 413	1473 ± 493	N/A	N/A
Iron (mg/day)	20 ± 6	20 ± 6	23 ± 6 ^a^	21 ± 6	17 ± 4	18 ± 7	N/A	N/A

* Significantly different from pre-program (*p* ≤ 0.05); ^a^ Significantly different from female (*p* ≤ 0.05); ^b^ Significantly different from heavy training (*p* ≤ 0.05).

**Table 4 nutrients-14-05203-t004:** Food group intakes of junior elite triathletes (mean ± SD) as per Australian Guide to Healthy Eating (AGHE) Guidelines.

Food Groups	Recommendations (Serves per Day)	Food Group Intake (Serves per Day)		Percent Meeting Age Appropriate AGHE Recommendations (%)	
		Pre-program	Post-program	Pre-program	Post-program
Fruit	≥2 (all ages)	1.9 ± 0.9	2.7 ± 1.5 *	50% (*n* = 9)	72% (*n* = 13)
Vegetable	≥5.5 (16–18 y) ≥5 (19 y+)	4.6 ± 1.8	4.6 ± 2.1	33% (*n* = 6)	33% (*n* = 6)
Grain	≥7 (16–18 y) ≥6 (19 y+)	7.9 ± 2.0	8.6 ± 2.5	72% (*n* = 13)	83% (*n* = 15)
Meat and alternatives	≥2.5 (16–18 y; 19 y+ female)≥3 (19 y+ male)	3.8 ± 1.1	3.3 ± 1.1	94% (*n* = 17)	77% (*n* = 14)
Milk and alternatives	≥3.5 (16–18 y) ≥2.5 (19 y+)	2.8 ± 1.3	3.5 ± 1.2 **	55% (*n* = 10)	61% (*n* = 11)
Discretionary	0–2.5 (16 y+ female) 0–5 (16–18 y male) 0–3 (19 y+ male)	3.8 ± 2.3	3.5 ± 2.0	39% (*n* = 7)	44% (*n* = 8)
Sports Products	N/A	0.7 ± 0.6	0.7 ± 0.8	N/A	N/A

* Significantly different from pre-program (*p* ≤ 0.05); ** Significantly different from pre-program (*p* ≤ 0.01).

## Data Availability

The data presented in this study are available upon reasonable request from the corresponding author.

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
