# Peer review of "The Impact of a ‘Remotely-Delivered’ Sports Nutrition Education Program on Dietary Intake and Nutrition Knowledge of Junior Elite Triathletes"

_nutrients, 2022, doi:10.3390/nu14245203_

Round 1

Reviewer 1 Report

The manuscript entitled "The Impact of a ‘Remotely-Delivered’ Sports Nutrition Education Program on Dietary Intake and Nutrition Knowledge of Junior Elite Triathletes" presents the results of an interesting study to assess the efficacy of a remotely delivered training program on nutrition knowledge and some diet parameters. Despite the topic is not completely novel, it remains interesting especially in the context to define the best practices and teaching methods.

Nevertheless, I feel that some suggestions could help to improve the quality of the manuscript to overcome some possibles weaknesses:

- The included sample in this study is small; this should be addressed in the limitations and might be worthy of including the definition of "pilot study" in the title and in the manuscript. This also recommends moderating a little the conclusions, as it looks too incisive for a study conducted in a small sample like the one from the current study. 

- Since the topic is not entirely novel, I suggest expanding the cited literature including also papers from other sports and nonsports contexts. Indeed, less than 30 references look too "concise" for such a topic; for example, there are recent manuscripts and reviews (including systematic reviews) that address this topic (e.g., Sánchez-Díaz et al., Nutrients, 2020; Foo et al., Nutrients; 2021).

- It is interesting the authors wanted to compare the data between males and females (please, check also the appropriate definition between "sex" and "gender", as strongly recommended in science also by the NIH - https://orwh.od.nih.gov/sex-gender#:~:text=%22Sex%22%20refers%20to%20biological%20differences,across%20societies%20and%20over%20time); maybe since they wanted to compare also the pre and post-program difference, a mixed-factors between-within ANOVA could give an additional insight over the results.

- Overall, in many parts of the manuscript, the authors explain that nutrition programs also refer to fluid intake and hydration; however, it seems that the authors did not consider this important factor in their analysis as it is well-known being one of the most overseen factors in nutrition and most athletes have poor hydration habits (Babiloni et al., Nutr Hosp, 2018; Buoite Stella et al., J Sports Med Phys Fitness, 2017; Suppiah et al., Nutrients, 2021). Can the author explain why some factors, e.g., fluid intake, USG or urine color were not considered? This should be better addressed in the discussion and highlighted in the limitations.

Author Response

Please see attached response to reviewer 1 comments.

Kind regards Greg

Reviewer 2 Report

I applause the initiative for the educational program, and for evaluating both knowledge and behavior. I find a few things in need to be addressed, to improve the understanding of setting, work, and results.

Who are the respondents: can information be provided about the program from which them are recruited? It seems difficult to understand why a single-arm study was chosen, considering the setting and opportunities to recruit a control group?!

Considering the one-arm non controlled setting, there are a few moves that can make findings more informative. First; why is no power calculation provided? Secondly, effect sizes could help us inform the impact of score-changes. Is a p-value of 0.05 suitable considering the many analytic procedures performed?

Results:

Line 168 (and more lines): why is “n” used to describe results? “n” usually describes a number of participants (and specifically logical when a percentage follows, as seen here), but a number of participants presented with decimals does not make sense. What do the number present?

Considering the aim of the program; to improve nutritional knowledge for triathletes specifically, it would be prudent to present results as recommended for athletes; i.e. gram macronutrient per kg body weight, and to comment on these findings. It seems like the proteinintake for females are pretty low, and

Table 2: what does the percentage denote?

Line 204 and more: I suggest a more interesting analysis would be to see if they better align with energy availability throughout the day, following the findings and highlights on withinday energy balance, by Torstveit et al 2018 and Fahrenholtz et al 2017.

Line 214; I suggest you present the recommendations in the method section too.

LINE 221 FINDINGS DESERVES COMMENTS IN DISCUSSION

What does “Discretionary” mean? (table 4)

Discussion and conclusion:

The limitation brough by the design of this study (single arm, non-controlled intervention) deserves consideration, and the conclusion needs to be softened. Specifically considering the young population, it may be reasonable to suggest that knowledge and behavior may increase/improve simply by maturation and experience.

What food sources brings the high iron intake? It is remarkably high among females, and a an evaluation on the bioavailability of the iron in the food would be informative.

Analysis that are controlled for or compared by previous individual nutrient guidance would be informative.

Author Response

Please see attached response to reviewer 2 comments.

Round 2

Reviewer 1 Report

I congratulate the authors for the appropriate responses given to my previous concerns. I recognize the efforts and also the limitations highlighted by the authors, as those regarding the sample size.

I have still one small comment I would like to ask the authors to address.
Regarding my last point, despite now the authors well-explained why they were not able to collect data about hydration status and fluid intake, I think that there is still space for a paragraph to discuss this important aspect of sports nutrition in the discussion, so even if the authors did not take any measurement in their study, they could refer to previous literature in order to provide the reader with some thoughts and take-home messages for further research and field application.

Otherwise, I feel like there is a disconnection between what the authors expressed in some parts of the text, and the lack of any discussion about it.

Author Response

Dear reviewer, 

Again, we thank you for your critical review of the manuscript. Your feedback has improved the over readability of the article and it's broader application to other settings. We have taken your additional feedback into consideration and further adjusted the manuscript. We have included the following paragraph in the manuscript and re-arranged the final paragraph of the discussion. I have included the tracked changed manuscript with these two sections highlighted for visibility. The following paragraph with associated references has now been included within the manuscript - lines: 275-288.

We devoted 10 of 83 questions within the sports nutrition knowledge questionnaire to hydration and included questions that assessed knowledge of beverage composition, implications of altered hydration status and appropriate fluid intake strategies for triathlon training specific scenarios. Ad libtium fluid intake during exercise is multifactorial [24] and likely influenced by external cues such as coach’s encouragement to drink [25] with athletes relying on performance support staff (i.e., athletic trainers) for hydration information [26]. While we carefully monitored dietary intake to assess daily energy and macronutrient intakes, we did not require participants to track daily fluid intake of non-calorie containing fluids and subsequently failed to measure daily total fluid intake before and after the education program. High sweat rates are commonly reported in endurance sport athletes [27], yet daily fluid intakes are poorly understood. Reporting total daily fluid intake and assessing hydration status would have added important additional context to our study and would be a worthy inclusion in future studies assessing changes in daily food and fluid intake in response to nutrition education interventions.

Additional references included:

Burke, L. M., Nutritional approaches to counter performance constraints in high-level sports competition. Exp. Physiol. 2021, 106 (12), 2304-2323.

Buoite Stella, A.; Francescato, M. P.; Sims, S. T.; Morrison, S. A., Fluid intake behavior in athletes during typical training bouts. J. Sports Med. Phys. Fitness 2017, 57 (11), 1504-1512.

Judge, L. W.; Kumley, R. F.; Bellar, D. M.; Pike, K. L.; Pierson, E. E.; Weidner, T.; Pearson, D.; Friesen, C. A., Hydration and Fluid Replacement Knowledge, Attitudes, Barriers, and Behaviors of NCAA Division 1 American Football Players. J. Strength Cond. Res. 2016, 30 (11), 2972-2978.

Barnes, K. A.; Anderson, M. L.; Stofan, J. R.; Dalrymple, K. J.; Reimel, A. J.; Roberts, T. J.; Randell, R. K.; Ungaro, C. T.; Baker, L. B., Normative data for sweating rate, sweat sodium concentration, and sweat sodium loss in athletes: An update and analysis by sport. J. Sports Sci. 2019, 37 (20), 2356-2366.

Kind regards Greg
